# Impact of Dentistry Materials on Chemical Remineralisation/Infiltration versus Salivary Remineralisation of Enamel—In Vitro Study

**DOI:** 10.3390/ma15207258

**Published:** 2022-10-17

**Authors:** Lia-Raluca Damian, Ramona Dumitrescu, Vlad Tiberiu Alexa, David Focht, Cristoph Schwartz, Octavia Balean, Daniela Jumanca, Diana Obistioiu, Dacian Lalescu, Sebastian-Aurelian Stefaniga, Adina Berbecea, Aurora Doris Fratila, Alexandra Denisa Scurtu, Atena Galuscan

**Affiliations:** 1Faculty of Dentistry, Department I, University of Medicine and Pharmacy “Victor Babes”, Eftimie Murgu Sq. No. 2, 300041 Timisoara, Romania; 2Translational and Experimental Clinical Research Centre in Oral Health, Department of Preventive, Community Dentistry and Oral Health, Faculty of Dentistry, University of Medicine and Pharmacy “Victor Babes”, Eftimie Murgu Sq. No. 2, 300041 Timisoara, Romania; 3Faculty of Veterinary Medicine, University of Life Sciences “King Michael I” from Timișoara, Calea Aradului No. 119, 300645 Timisoara, Romania; 4Faculty of Food Engineering, University of Life Sciences “King Michael I” from Timișoara, Calea Aradului No. 119, 300645 Timisoara, Romania; 5Computer Science Department, West University of Timisoara, UVT, 300223 Timisoara, Romania; 6Faculty of Agriculture, University of Life Sciences “King Michael I” from Timișoara, Calea Aradului No. 119, 300641 Timisoara, Romania; 7Faculty of Dental Medicine, Ludwig-Maximilian University Munich, Goethestr. 70, 80336 Munich, Germany; 8Research Centre for Pharmaco-Toxicological Evaluation, University of Medicine and Pharmacy “Victor Babes”, Eftimie Murgu Square No. 2, 300041 Timisoara, Romania

**Keywords:** confocal microscopy, remineralisation, salivary pH, resin infiltration, Fluor, prevention dentistry, atomic absorption spectroscopy, macro elements

## Abstract

The aim of this study is to evaluate salivary remineralisation versus chemical remineralisation/infiltration of enamel, using different dentistry materials. The enamel changes were studied using confocal laser scanning microscopy (CLSM), and the depth of lesions and demineralisation/remineralisation/infiltration percentage were calculated. Additionally, the macro elemental composition of the teeth was performed using atomic absorption spectroscopy (AAS). Two studies were performed: (i) demineralisation of enamel in 3% citric acid and infiltration treatment with infiltration resin (Icon, DMG), remineralisation with Fluor Protector (Ivoclar Vivadent) and artificial saliva pH 8; and (ii) enamel demineralisation in saliva at pH 3 and remineralisation at salivary pH 8. The results showed that, firstly, for the remineralisation of demineralised enamel samples, Fluor Protector (Ivoclar Vivadent) was very effective for medium demineralised lesions followed by saliva remineralisation. In cases of deep demineralisation lesions where fluoride could not penetrate, low viscosity resin (Icon, DMG, Hamburg) effectively infiltrated to stop the demineralisation process. Secondly, remineralisation in salivary conditions needed supplementary study over a longer period, to analyse the habits, diet and nutrition of patients in detail. Finally, demineralisation/remineralisation processes were found to influence the macro elemental composition of enamel demineralisation, with natural saliva proving to be less aggressive in terms of decreasing Ca and Mg content.

## 1. Introduction

A primary strategy for the effective long-term prevention of enamel demineralisation and carious lesions is essential for understanding the manipulation of the oral environment in terms of salivary parameters, ion flows, pH and oral microbiome [1].

Over time, studies in diet and nutrition have shown that consuming carbohydrates and acids leads to carious lesions. In this process, saliva plays a significant role, particularly, salivary pH. Saliva has an optimal pH between 6.7 and 7.4, but through prolonged exposure to acids and carbohydrates in the oral cavity, bacteria break down carbohydrates releasing lactic acid and aspartic acid, which lead to lower saliva pH values [2,3]. When the salivary pH falls below the critical value of 5.5, acids start to decompose the enamel [1,3,4,5]. The more often that the enamel is periodically exposed to these conditions, the more it starts losing minerals from a layer that stretches a few micrometres into the subsurface, in a process called softening. When affecting larger surfaces of the enamel, this softening process can lead to demineralisation, and even a loss of surface enamel [4,5,6,7]. In the case of failed dental plaque removal, a frequent intake of carbohydrates will tip the dynamic balance between demineralisation and remineralisation in favour of demineralisation, accompanied by chalky white spots [3]. Enamel demineralisation is based on the fact that the main driving force is the transport of hydrogen ions from the dental plaque at a pH of 5 to the underlying enamel at a pH of 7 [3]. Saliva has an essential role during the intermediate periods, between meals, when remineralisation can naturally occur, as it can neutralise acid, provide mineral ions and form a protective film for the enamel [1,3,6,7]. In recent years, non-invasive remineralisation methods have been introduced based on the application of fluoride-based remineralising agents. Fluoride interacts with oral fluids on the enamel surface, reacting with calcium and phosphorus ions to form fluorapatite [8].

However, the benefits depend on the frequency of application and concentration [9]. Nevertheless, there is a layer of hyper-mineralised surface enamel in all incipient demineralisation lesions, which does not allow the penetration of chemical remineralising agents very easily. In the past decade, researchers have introduced a technique involving the infiltration of resins with low viscosity, a new minimally invasive concept [10]. Infiltrating resin Icon (DMG, Hamburg, Germany) has shown promising results in previous studies. The manufacturer’s 15% hydrochloric acid gel effectively removes the surface layer of demineralisation lesions, after which, the Icon Dry ethanol removes the moisture inside the microporosities. These two stages allow the resin to penetrate the lesion’s capillary forces and occlude the lesioned surface’s pores [11]. Currently, the long-term stability and possible surface-modifying effects of resin are being studied. The degradation of the resin in time may lead to surface problems, such as the appearance of plaque on these surfaces and, therefore, the development of secondary caries [12,13].

Among the various devices used to analyse the treatment of demineralisation lesions, confocal laser scanning microscopy (CLSM) is considered a valuable tool in dental research. This type of microscope is an optical imaging technique which overcomes the specific limitation of traditional fluorescence microscopy. The resolution improvement, and removal of ambiguities outside the focus, allow for more information to be obtained through the three-dimensional reconstruction of the sample. Its applications include researching tooth enamel, tooth decay, the response of hard and soft tissues to biomaterials (e.g., implants), monitoring the effect of periodontal treatment regimens, and analysing the penetration of fluid materials into the depth of tooth enamel [14,15].

Some studies have evaluated the penetration of infiltrating resins and the penetration of Fluor Protector. However, our study aims to emphasise the importance of the results for the clinical practice of the dentist. In clinical practice, since many cases of lesions of enamel demineralisation are encountered, it is very important that the dentist, after a correct diagnosis of the lesion, chooses the therapy suitable for each case, as follows:For incipient enamel lesions, the only recommendation is to modify the diet to increase the salivary pH, for remineralisation to occur as a natural process with the help of saliva;For a medium demineralisation lesion, the dentist can choose a treatment based on fluoride solutions; the studies show that fluoride therapy has very good results;For enamel demineralisation lesions (without loss of substance), remineralising fluoride therapy is no longer sufficient; in this case, it is necessary to infiltrate the enamel with low viscosity resin to stop the demineralisation process by obliteration of the pores. The diagnosis of demineralisation can be accurately performed in clinical practice with the help of a DiagnoDent laser diode.

This study includes two protocols of demineralisation because, in the oral cavity, several types of agents produce a demineralisation lesion: carbonated drinks, carbohydrate consumption, different acid gels applied on enamel, gastroesophageal reflux disease, and acid pH of saliva. For our study, we chose citric acid, with a pH similar to carbonated drinks, which has a short but aggressive demineralisation action. We chose a demineralisation protocol with acid saliva, which can simulate oesophageal reflux disease or the salivary acid pH of a patient with a balanced diet, which is not as aggressive on enamel but can cause continuous damage. In the established protocol, the chosen variables were the periods selected (24 and 48 h).

Therefore, our study aimed to assess the potential for infiltration of chemical and natural substances into demineralisation lesions in human tooth enamel, with confocal laser microscopy. The content of mean macro elements (Ca, Mg, K) in the teeth before and after demineralisation in citric acid and the impact of natural remineralisation were also studied.

## 2. Materials and Methods

The research included two directions.

Study 1: Chemical demineralisation versus chemical infiltration. The first study analysed the demineralisation of human enamel in citric acid at 3% concentration, being the equivalent of a pH of 3.5, at different periods (3 min, 5 min, 7 min) to see at what point in time the demineralisation was initiated, and to what depth it reached. After that, the samples were subjected to chemical infiltration (with Fluor Protector, Ivoclar Vivadent; and Icon, DMG, Hamburg) and pH 8 artificial saliva to observe the penetration capacity of the substances. Icon was used in the study to show that when fluoride no longer penetrates sufficiently to produce remineralisation of the entire lesion, and to prevent the transformation of a demineralisation lesion into a carious lesion, there is an option of infiltration therapy using a low viscosity resin that penetrates to a greater depth.

Study 2: Salivary demineralisation versus Salivary remineralisation. This study analysed the demineralisation of human enamel samples in acidic artificial saliva, pH 3, and the penetration capacity of basic saliva, pH 8, into the demineralised lesions. The artificial saliva was able to simulate the actual situation in the human mouth because microorganisms in the oral cavity have no part in the mineralisation of the tooth. Once maturity has been reached, mineralisation is ended, and microorganisms play a greater role in plaque-forming or biofilm appearance on the tooth surface [16,17].

Saliva was used in this study only in terms of pH+, with our interest in its acid–base properties, not in its microorganism content. In vivo, pH triggers the process of demineralisation/remineralisation in the oral cavity by migrating ions from the enamel into the saliva, and vice versa.

A diagrammatic presentation of the entire protocol study is presented in Figure 1.

### 2.1. Biological Material

A review of the literature showed that: Arnold et al. [18] used 12 extracted caries-free human incisors; Péreza et al. [18] used 24 fluorosed human molars and premolars; Elsami B. et al. [19] used 15 patients who needed 2 premolars extracted; Meyer-Luekel et al. [20] used 20 extracted permanent human posterior teeth with non cavitated caries; Arnold et al. [12] used 28 permanent teeth extracted with non cavitated caries lesions; Enan et al. [21] used 45 extracted premolars, resulting in 90 specimens; Chokshi et al. [22] used 60 enamel specimens; and Zhang et al. [23] used 50 artificial enamel white spots. Therefore, we considered that our number of 60 samples from 20 human teeth, third permanent molars from patients older than 18 years, was relevant.

The selection criteria were based on the fact that once maturity has been reached, the mineralisation of an adult tooth has ended, and the age difference is no longer important. Gender also does not affect the results [24], since the comparison was performed between the same samples divided for different activities within our research. Therefore, orthodontically extracted teeth from patients over 18 years of age who had not been subjected to any influences, were chosen for this study. The standardised mineralisation of teeth was not measured, as the average demineralisation of each sample was used as a starting point.

The research analysed 20 human teeth, with 60 enamel sections taken from unerupted wisdom teeth unaffected by caries. The sections were cut at dimensions of 7 mm-10 mm-5 mm, in the vestibule–oral direction and kept in a solution of 0.9% NaCl until the analysis. For this study, the patient’s consent was signed for the extracted tooth to be analysed in the research. Additionally, ethical approval was received from the Scientific Research Ethics Committee of the Victor Babes University of Medicine and Pharmacy Timisoara, Romania, registered under number 45/28.09.2018.

### 2.2. Confocal Laser System Microscopy Analysis (CLSM)

Previous studies have used SEM for enamel analysis, but evaluated the enamel’s surface area and microhardness. Therefore, we considered it opportune to determine the depth at which demineralisation occurs and the best method for remineralisation, for which we proposed confocal microscope analysis. The enamel samples were analysed using a Leica TCS SPE- Leica Microsystem, according to the method used in [25]. The TCS SPE confocal microscope contains up to four lasers in the solid state: 488 nm, 532 nm, 635 nm, these being standard arousal lines; and a 405 nm laser—optional—for nuclear colouring. Lasers have a significant advantage in that they are durable and do not require external cooling. The Leica TCS SPE is the only confocal device with accurate spectral detection in its class. It is based on a prism, which spreads light in its spectrum on a device for detecting the spectral domain, so the system allows a freely adjustable spectral detection from 430 to 750 nm. The enamel sections were dried and coloured for each study with 1% fluorescein alcoholic solution (NaFl, 518-47-8, Sigma Aldrich, Merck KgaA, Darmstadt, Germany) for 3 min and subsequently washed in deionised water for 10 s. The images were analysed using a 10× objective in dual fluorescence mode.

### 2.3. Protocol Study 1

#### 2.3.1. In Vitro Chemical Demineralisation

For this study, 35 enamel samples were used: 5 control samples, 10 treated with fluoride, 10 treated with infiltrated resins, and 10 samples remineralised in basic artificial saliva (Figure 1). As the enamel samples were taken from unerupted, intact molars that had not been subjected to any abrasion in the oral cavity, we tried to simulate, as closely as possible, the environment in the oral cavity. Therefore, all enamel samples were subjected to microabrasion, for which a Lex XT 9 mm-3 M Soft disc and elongated cone grinding gums of fine granulation Nais 2002, were used. Except for the control samples, the enamel samples were immersed in 25 mL of citric acid at a concentration of 3% (Sigma Aldrich), the equivalent of pH 3.5, to determine a chemical demineralisation lesion in vitro. Immersion in citric acid was achieved in three cyclic stages: 3 min–5 min–7 min, according to the protocol presented in Table 1. The lesions produced at different penetration depths (P.D.) between 15 and 45 µm were observed and measured.

The samples were analysed through CLSM after each immersion time, resulting in demineralisation absent at 3 min, initiated at 5 min, and present in the third immersion cycle.

#### 2.3.2. Infiltration of Demineralised Lesions with Fluor Protector (Fluor Protector, Ivoclar Vivadent)

Ten in vitro demineralisation samples were subjected to treatment with Fluor. Fluor Protector ampoule 1 mL, by Ivoclar Vivadent, was used for the study, containing 0.1% fluoride in a homogeneous solution. After washing in distilled water and drying, the fluoride protective varnish was applied to the demineralised surface and air dried for 10 min. After drying, the samples were analysed through CLSM.

#### 2.3.3. Infiltration of Demineralisation Lesions with Icon, Low Viscosity Resin (Icon, DMG, Hamburg)

Another 10 demineralised in vitro samples were treated with a combination of infiltrating resins termed Icon (DMG, Hamburg), using the following protocol. The samples were washed with distilled water for 30 s, then Icon Etch (hydrochloric acid 15%, pH 1) was applied for 2 min and washed again for 30 s with distilled water. Icon Etch was applied for a further 1 min and then washed for 30 s with distilled water. After that, Icon Dry (ethyl alcohol, pH neutral) was applied for 30 s. The next step was the application of Icon Infiltrator for 3 min and photopolymerisation for 40 s. The process with Icon Infiltrator was repeated for 1 min and photopolymerised again for 40 s. For photopolymerisation, a Photo Lamp Wireless LED E Woodpecker, 100 V–240 V.s, was used. After applying the infiltrated resin, the enamel samples were stained with a 1% alcoholic solution of fluorescein and analysed with a confocal microscope.

#### 2.3.4. Infiltration of Demineralisation Lesion with Artificial Saliva of Basic pH 8

The last ten samples of demineralised enamel were put in 25 mL of artificial saliva with basic pH of 8.5 in four cycles: 12 h, 24 h, 48 h and 72 h, at room temperature. After each immersion cycle, the samples were stained with 1% fluorescein alcoholic solution and were analysed under the CLSM.

Artificial saliva was prepared at the Victor Babes Faculty of Pharmacy for this study.

##### Preparation of the Artificial Saliva

The reagents required for the preparation of artificial saliva were: CaCl_2_ ·2H_2_O (purity > 99.5%) from Honeywell Fluka™ (Charlotte, NC, U.S.A.); NaCl (purity > 99.5%) from Chimopar S.A (Bucharest, Romania); CO(NH_2_)_2_ (purity > 99.5%) from Sigma-Aldrich (St. Louis, MO, USA); KCl (purity > 99.8%) and NaOH pellets (purity > 99.3%) from Chimreactiv (Bucharest, Romania); and HCl 37% from Honeywell Fluka™ (Charlotte, NC, USA).

Artificial saliva solutions with different pHs were prepared as follows: four solutions with acidic pH (pH 3 and 5) and one with basic pH (pH 8), according to the method described by Dinu et al. [26].

Initially, pH-neutral saliva was prepared by dissolving 0.40 mg/L NaCl, 0.40 mg/L KCl, 0.80 mg/L CaCl_2_ × 2H_2_O, and 1 mg/L CO(NH_2_)_2_ in distilled water until a clear solution was obtained. Then, the saliva solutions with acidic pH were obtained by treating the neutral saliva with 37% HCl until values of pH 3 and pH 5 were achieved. Finally, to obtain the artificial saliva with a pH value of 8, 10 N NaOH was added over the initially prepared saliva. The pH of the artificial saliva was measured using a Thermo Scientific Eutech pH 150 electrode pH meter (Thermo Scientific, Waltham, MA, USA).

### 2.4. Protocol Study 2: Salivary Demineralisation and Infiltration with Artificial Saliva

Twenty-five enamel samples were used, being sliced sections of unerupted wisdom teeth unaffected by caries and artificial saliva (obtained according to the protocol described in study 1). All enamel samples were subjected to microabrasion, for which a Lex XT 9 mm-3 M soft disc, and elongated cone grinding gums of fine granulation Nais 2002, were used. The microabrasion protocol was used because the enamel samples were taken from unerupted, intact molars that had not been subjected to any abrasion in the oral cavity. For this study, 5 enamel samples were kept blank, 10 samples were introduced into 25 mL of artificial acid saliva of pH 3, which we coded as N, and 10 samples were introduced into pH 5, which we coded as Z, in three cycles: 12 h, 24 h, 48 h, to create the in vitro demineralisation lesion. After each cycle, the samples were washed in distilled water, dried, and coloured with 1% fluorescein alcoholic solution for 3 min and 10 s in distilled water, according to the protocol, and analysed through CLSM [25].

This study aimed to remineralise natural lesions in saliva with pH 8. The demineralised enamel samples, after measurement, were each introduced into 25 mL basic artificial saliva of pH 8, initially in three cycles: 12 h, 24 h, and 48 h. After each cycle, the samples were analysed by confocal microscopy, where we found that at 48 h, traces of remineralisation began to be visible only at the level of the Z enamel samples, and were demineralised in pH 5. We decided to reintroduce the samples for up to 72 h in saliva. After 72 h, the samples were washed with distilled water, stained with 1% alcoholic fluorescein solution, and analysed using CLSM.

### 2.5. Macro Elemental Composition

The macro elements (Ca, K, Mg) were detected using atomic absorption spectroscopy (AAS) after the acidic dissolution of the organic matrix. In this regard, 3 g of sample was dissolved in 6 N HCl (Sigma-Aldrich Chemie GmbH, München, Germany), and kept overnight in an acidic solution. After filtration, the solution was brought to a volume of 50 mL, with distilled water. The metal content was determined by atomic absorption spectrometry technique, using a Varian Spectra 240 FS spectrophotometer (Palo Alto, CA, USA), using the following protocol: an air:acetylene ratio of 13.50:2, nebuliser uptake rate of 5 L/min, and working solutions with concentrations ranging from 0.3 to 3 μg/L, prepared from multi-element ICP standard solution 1000 mg/L.

The samples were divided into equal parts to establish a relationship between the remineralisation protocols and the macro elemental composition. To better evaluate the mineral content in the region submitted to the different remineralisation protocols, we submerged the entire sample surface according to protocol, therefore, not having lesions smaller than the entire surface taken into analysis. Using AAS, the entire sample was analysed, and the data were used for statistical analysis. The results were expressed as the value obtained divided by the mass taken into analysis before acid immersion, so the values obtained were easily extrapolated.

### 2.6. Statistical Analysis

All replicates’ mean values and standard deviations were calculated using GraphPad Prism (v.5.0 software, Manufacture, San Diego, CA, USA). Differences between means were analysed with one-way ANOVA, followed by multiple comparisons using the Duncan test (two-sample assuming equal variances). Differences were considered significant when *p*-values < 0.05. Correlations between variables were performed using Microsoft Excel 2010. Duncan tests, linear correlations between variables, principal components analysis and cluster analyses were performed using Statistical (v.12, TIBCO Software, Palo Alto, CA, USA).

## 3. Results

### 3.1. Chemical Demineralisation and Infiltration with Fluor Protect (Ivoclar Vivadent)

Initially, as a preliminary analysis, in accordance with [12], intact enamel samples were analysed without undergoing any demineralisation process; subsequently, the samples were immersed in citric acid and then re-analysed, where differences in enamel structure could be observed.

Figure 2 presents the CLSM images (10× objective in dual fluorescence mode) of enamel after demineralisation/remineralisation with chemical and natural agents, and Table 2 shows the values obtained regarding the dimensions of lesions before and after treatment.

Figure 2a presents the aspect of enamel after demineralisation with 3% citric acid, while Figure 2b,d present the CLSM profile after Fluor Protector infiltration of enamel in different stages of penetration. It was observed that fluoride’s infiltration capacity differs depending on the lesion’s depth. In the case of superficial lesions, the penetration capacity was maximum (100%, Figure 2b), while for deep lesions between 95 and 120 µm, the penetration capacity of fluoride was partial (Figure 2c). Figure 2d presents the aspect of enamel after Fluor Protector infiltration with partial penetration, where the demineralisation lesions can be observed.

Figure 2e,f present images recorded with a Leica confocal microscope in respect of treatment with infiltration resin Icon (DMG, Hamburg). It can be seen that a full infiltration of lesions was obtained after demineralisation with 3% citric acid.

Figure 2g shows the difference between the healthy enamel and the surface of the enamel treated with infiltration by Icon resin. Figure 2h shows the laser penetration plot where light penetrates the healthy enamel, but is blocked in the enamel treated with infiltrated Icon resin.

Figure 2i shows images of enamel demineralised in citric acid, and subsequently immersed in pH 8 basic artificial saliva. CLSM captured the initiation of enamel remineralisation in saliva, and in Figure 2j, a sound enamel surface is shown.

The results presented in Table 2 show that before Fluor Protector (Ivoclar Vivadent), the values recorded for the lesions of demineralised enamel samples were between 87 μm and 143 μm, with an average of 115 μm, and decreased at 77 μm (minimum value 64 μm and maximum value 81 μm, respectively). The demineralisation lesion depth measured before fluoride therapy had a depth average of 115 μm, which after fluoride therapy decreased to 77 μm, indicating that remineralisation of a portion of the lesion occurred. Correlating with Figure 2b–d, it can be stated that the degree of penetration is dependent on the depth of the lesion and the degree of demineralisation.

The results of the samples demineralised in citric acid, on which infiltration with the low viscosity resin Icon (DMG, Hamburg) was applied (Figure 2e,f), showed that the average penetration depth of the demineralised enamel samples was 115.60 μm, and the average penetration depth after Icon treatment was 115.60 μm; the minimum and maximum penetration depths of the demineralised enamel samples were 90 μm and 141 μm, respectively, and after treatment with Icon were 90 μm and 141 μm, respectively, which meant that Icon penetrated the lesion completely.

Figure 2i,j present images of the initiated pH 8 saliva remineralization, and the sound enamel samples. In this case, the penetration depth value recorded for the demineralised sample was 118.81 μm, and decreased to an average value of 39.90 μm after treatment with basic saliva. The minimum and maximum penetration depths of the demineralised enamel samples were 94 μm and 145 μm, respectively, and following treatment with basic saliva, the penetration depths were 30 μm and 45 μm, respectively, suggesting a partial penetration of the lesion by the mineralising agent.

Figure 3 presents citric acid enamel demineralisation percentages, calculated according to the percentage formula, after 3 min (Figure 3a), 5 min (Figure 3b) and 7 min (Figure 3c) of exposure.
(1)$$p=\frac{a}{b}\times 100$$
where *a* is the number of samples meeting a certain condition, and *b* is the total number of samples.

Figure 3a presents the percentages of demineralisation in 3% citric acid for three minutes at various penetration depths. Notably, two enamel samples (representing 6.67%) had demineralisation values between 10 μm and 15 μm depth, eight enamel samples (representing 26.67%) had demineralisation values between 15 μm and 20.7 μm depth, and enamel samples (representing 23.33%) had demineralisation values between 20 μm and 25 μm depth. Additionally, two enamel samples (representing 6.67%) had demineralisation values between 25 μm and 30 μm depth, three enamel samples (representing 10%) had demineralisation values between 30 μm and 35 μm depth, four enamel samples (representing 13.33%) had demineralisation values between 35 μm and 40 μm depth, and the last four enamel samples (representing 13.33%) had demineralisation values between 40 μm and 45 μm depth.

Regarding the samples of enamel demineralised in 3% citric acid for 5 min, we can summarise from Figure 3b that the degree of demineralisation increased and the percentage of injuries caused at great depths increased. Eight samples of enamel (representing 26.67%) had demineralisation values between 30 μm and 40 μm depth, and four enamel samples (representing 13.33%) had demineralisation values between 40 μm and 50 μm depth. Lesions at a depth between 50 μm and 60 μm were observed in 16.67% of samples. An amount of 23.33% of samples produced a lesion depth between 60 μm and 70 μm, and six enamel samples (representing 20%) had demineralisation values between 70 μm and 80 μm.

For the enamel samples demineralised in 3% citric acid for 7 min, Figure 3c shows the increment of lesion depths. Therefore, 10% of enamel samples had demineralisation values between 80 μm and 90 μm, 26.66% of enamel samples had demineralisation values between 90 μm and 100 μm, while 10% of enamel samples had demineralisation values between 100 μm and 110 μm. The degree of demineralisation increased until 150 μm, namely, 3.33% of samples of enamel had demineralisation between 110 μm and 120 μm, 10% of samples were demineralised in the interval 120 μm and 130 μm, 30% between 130 μm and 140 μm, and another 10% of enamel samples had demineralisation values between 140 μm and 150 μm.

In the case of enamel samples demineralised in 3% citric acid for 3 min, the average value of the demineralised enamel samples was 26.56 μm. The minimum value of the demineralised enamel samples was 14 μm; the maximum value of the demineralised enamel samples was 43 μm.

In the case of enamel samples demineralised in 3% citric acid for 5 min, it was observed that the interval of lesions produced was in the range of 37–77 μm, with the average value comparatively increasing to 55.20 μm. The standard deviation was defined as 14.14 μm.

When the demineralisation of enamel samples was performed in 3% citric acid for 7 min, we noticed that the lesion size varied between 87 and 145 μm, and the average value of the demineralised enamel samples was 116 μm.

In order to study the statistically significant differences between the distributions of demineralisation depths at the three different cycles of time studied, the Duncan test for multiple comparisons was used (Figure 4). Significant statistical differences were observed between the depth of enamel samples demineralisations at 3 min, and the depth of enamel samples demineralisations at 5 min. Additionally, differences were observed between the depth of demineralisations of enamel at 3 and 7 min, and between the depth of enamel sample demineralisations at 5 and 7 min, respectively.

Figure 5 presents the Duncan test applied to the three studied treatments representing, respectively, the depth of lesions after demineralisation, and the depth of penetration of the remineralisation agent. The analysis highlighted a significant statistical difference between the samples before and after treatment with Fluor Protector (Ivoclar Vivadent), and between the demineralised sample before application of saliva pH 8 and after remineralisation with saliva. Conversely, no statistically significant differences were observed between the Icon-demineralised sample and the sample with Icon treatment.

It is noteworthy that following the treatment with Fluor Protector (Ivoclar Vivadent), Icon (DMG, Hamburg) and basic artificial saliva pH 8, the results revealed statistical differences between all samples remineralised with different agents. Therefore, there was a significant statistical difference between the pairs of remineralised samples used: Fluor Protector (Ivoclar Vivadent)/Icon (DMG, Hamburg, Germany); Fluor Protector (Ivoclar Vivadent)/artificial saliva pH 8; and Icon (DMG, Hamburg, Germany)/artificial saliva. These differences suggested different degrees of penetration, and effectiveness, of the three remineralisation agents studied.

Of the three treatments applied to the samples of enamel demineralised in citric acid, the most effective was the treatment with low viscosity infiltrating resin Icon, from DMG, Hamburg, which managed to fully penetrate the demineralisation lesion. Fluoridation also further proved its importance, with the results of fluorine treatment with Ivoclar Vivadent being excellent, but unable to equal the infiltrating resin. Remineralisation of samples in artificial saliva occurred, which was very important, but it was considered necessary to further study remineralisation over a more extended period. Our results showed that in terms of remineralisation, fluoride had a limited ability to penetrate in depth, and if the lesion was much deeper, a demineralised area would remain below the depth penetrated by fluoride. On the other hand, Icon penetrated deep lesions, obliterating the pores. The data resulting from our study may lead to a comparison of efficacy. A comparison is important so a dentist knows how to optimally choose which therapy to apply to the demineralisation lesion in clinical practice: either, remineralisation with fluoride and increase of the salivary pH, or infiltration therapy with low viscosity resin.

### 3.2. Salivary Demineralisation and Infiltration

For this study, 20 enamel samples were subjected to demineralisation in artificial saliva: 10 samples were introduced into artificial saliva with pH 3, and 10 were introduced into artificial saliva with pH 5.

#### 3.2.1. Demineralisation in Salivary Acid Medium

After CLSM analysis of the surfaces of the enamel samples (Figure 6), it was observed that samples introduced into artificial saliva with pH 3 for 48 h showed incipient demineralisation lesions at 12 h after immersion, medium lesions at 24 h after immersion, and presented evident lesions more extensively in the surface, at 48 h. The samples were microscopically compared with hard, healthy enamel sections taken from the same dental units.

CLSM analysis of the enamel samples inserted into artificial saliva with pH 5 showed that demineralisation, in this case, was present 48 h after immersion (Figure 7). The samples were microscopically compared with hard, healthy enamel sections taken from the same dental units.

In order to study the statistically significant differences between the distributions of demineralisations of the enamel samples immersed in artificial saliva with pH 3 at 12 h, 24 h, and 48 h, the Duncan test was used for multiple comparisons. Statistically significant differences were found between all samples immersed in artificial saliva with pH 3 (Figure 8). Regarding the enamel samples demineralised in artificial saliva with pH 5, it was observed that between the demineralisation of the enamel samples immersed in artificial saliva at 12 h and the demineralisation of the enamel samples immersed in artificial saliva at 24 h, no statistically significant differences were recorded. However, statistically significant differences were revealed between the demineralisation of the enamel samples immersed in artificial saliva at 12 h and 24 h, and the samples immersed in artificial saliva at 48 h.

There were statistically significant differences recorded between the demineralisation of the enamel samples immersed in artificial saliva with pH 3 and pH 5, at 12 h, 24 h and 48 h, respectively (Figure 8).

#### 3.2.2. Remineralisation in Basic Salivary Medium

According to the procedure described in Section 2.4, demineralised enamel samples were individually introduced into 25 mL of basic artificial saliva of pH 8, initially in four cycles: 12 h, 24 h, 48 h, and 72 h. The results observed after examination with CLSM, in the case of enamel samples demineralised in artificial saliva at pH 3, were that remineralisation was present at 72 h, while in the case of demineralisation samples in pH 5, remineralisation was initiated at 24 h (within three samples), initiated in the case of all samples at 48 h, and present in the case of all samples at 72 h (Figure 9).

There were statistically significant differences between the remineralisation of N samples (i.e., demineralised in saliva with pH 3) at 12 h, 24 h, 48 h and 72 h. However, no statistically significant differences were found between the remineralisation of the N samples at 12 h and 24 h, at 12 h and 48 h, and at 24 h and 48 h (Figure 9).

Regarding the remineralisation of the Z samples (i.e., demineralised in saliva with pH 5), there were statistically significant differences recorded between 12 h and 24 h, between 12 h and 48 h, and between 12 h and 72 h (Figure 10).

Additionally, statistically significant differences were noted between remineralisations of Z samples at 24 h and 48 h, between remineralisations of Z samples at 24 h and 72 h, and between remineralisations of Z samples at 48 h and 72 h.

For differences between remineralisations of N and Z samples at the same time interval, the following results were recorded: there were no statistically significant differences at 12 h, but there were statistically significant differences at 24 h, 48 h and 72 h.

From the statistical analysis of study 2, it was concluded that demineralisation in acid saliva pH 3 was initiated much earlier than in saliva pH 5—as early as 12 h in the case of some probes—and remineralisation, instead, was initiated faster in the case of demineralised samples in saliva pH 5.

### 3.3. Macro Elemental Composition

The Ca content (Figure 11a) decreased slightly in the samples after demineralisation. The decrease was 0.54% in the samples demineralised with 3% citric acid and evaluated after 72 h. The remineralisation led to an increase in Ca content compared not only with the control but also the demineralised sample, the increase being more pronounced in the case of salivary remineralisation with pH 8 saliva (8.64% in relation to the control, and 9.23% in relation to the demineralised sample), compared with remineralisation in a naturally acidic environment (5.40% in relation to the control, and 6.28% in relation to the demineralised sample).

Figure 11a demonstrates that there were no statistically significant differences between the Ca content of enamel samples demineralised with 3% citric acid and the Ca content of enamel samples demineralised with saliva pH 3, with, respectively, the Ca content of enamel samples remineralised with saliva pH 8. Additionally, no statistically significant difference existed between the Ca content of demineralised/remineralised enamel samples and the control enamel samples.

In Figure 11b, there were no statistically significant differences between the Mg content of enamel samples demineralised with 3% citric acid and the Mg content of enamel samples demineralised with saliva pH 3, with, respectively, the Mg content of enamel samples remineralised with saliva pH 8. There were no statistically significant differences between the Mg content of the demineralised/remineralised enamel samples and the control enamel samples.

In Figure 11c, there were no statistical differences between the K content of enamel demineralised with 3% citric acid and the K content of enamel samples remineralised with saliva pH 8. Additionally, there were no statistically significant differences between the K content of demineralised enamel samples with saliva, and, respectively, the K content of enamel samples remineralised with saliva pH 8, and control enamel samples. It was observed that there were statistical differences between the K content of enamel samples demineralised with 3% citric acid, and the K content of enamel samples demineralised with saliva pH 8, and control enamel samples.

The Mg content decreased by 1.26% after demineralization, in relation to the control. Remineralisation in a basic environment did not increase Mg content compared to the values recorded in the control or the demineralised sample, and remineralisation in an acidic natural environment led to an insignificant increase in relation to the demineralised version (1.76%).

The K content increased by 16.06% as a result of demineralisation, compared with the control, and the salivary remineralisation in a basic or acidic environment led to a decrease of 11.33% in the K content, in relation to the demineralised sample. An increase of 5.64% occurred in the case of remineralisation at basic pH, compared with the control, and a decrease of 3.33% in the case of remineralisation in an acidic environment.

## 4. Discussion

### 4.1. Chemical Demineralisation/Remineralisation

To our knowledge, this is the first study that used infiltrator resin (Icon, DMG, Hamburg, Germany) and Fluor Protector (Ivoclar Vivadent) for chemical remineralisation. With the help of CLSM analysis, we measured the depth of the demineralisation lesions to observe the resin’s penetration, versus the fluoridating agent’s penetration. The results of the measurements validated the importance of fluorine treatment because it obtained good results in the remineralisation process; while it failed to penetrate the depth of the lesion, it covered the surface layers (78–81 μm). Similar results in terms of remineralisation with fluoridating agents were found in the study conducted by Farhadian et al. [26], where a single dose of fluoride with a high concentration produced a significant, but not total, reduction in the demineralisation lesion from 45–62 μm to 37.3 ± 10.8 [26]. Following this, other authors observed that a high concentration in a single dose was not important, but in order to prevent demineralisation and favour remineralisation, the frequency of fluoride application was more important [27]. Another study demonstrated that application of high-concentration fluoride caused the rapid precipitation of minerals from the enamel surface and the obturation of the enamel pores that connect with the underlying demineralised lesion [28,29].

Simulations of molecular dynamics of fluoride incorporation into hydroxyapatite showed that fluoride ions are easily incorporated from the solution into the hydroxyapatite surface, but does not separate, and remains on the surface [30]. That study was very important because it helps us understand the role of fluoride in stabilising hydroxyapatite in tooth enamel, and the study concluded that only repeated exposure to fluoride will create a lasting effect on the structure of the enamel [30].

This year, a comprehensive study of the literature regarding the available dates on the interaction between fluoride and enamel was published [31]. According to that study, the amount of fluoride incorporated into tooth enamel was very small, a few μg/mm^2^, and was unlikely to protect the tooth against acid attack, be it an acid agent or acid-producing cariogenic bacteria [31].

Instead, there were numerous literature studies in which the technique of infiltration of a low viscosity resin was investigated as a new concept in minimally invasive dentistry, and which exceeded the results of other demineralising agents [11]. These low viscosity resins exclude the pores of the demineralising injurious body by capillary forces [11].

Compared to fluoride, where our study showed penetration of up to 80 μm into the depth of the lesion, there are literature studies that demonstrate, in the case of Icon infiltrating resins, that only demineralisation cycles with HCl produce a demineralisation of up to 77.56 μm [32].

Our study showed that the penetration of Icon resins infiltrating through capillary forces succeeded up to a depth of 141 μm; all demineralisation lesions we measured and treated with Icon resin were fully penetrated. The potential of resins to penetrate completely, or almost completely, into the demineralisation lesion was also proven in other literature studies. Cazzolla et al. reported similar results to those obtained in our study, where it was concluded that low viscosity infiltration resins almost penetrated the enamel lesions [32]. Infiltrating resins can completely seal the demineralisation lesions; therefore, they can be used as a fast and effective treatment [33,34,35]. In a study by Subramaniam et al., infiltrating resin Icon was shown to have penetrated to the depth of 276 μm [34]. Recent advances regarding both procedural and technological devices, which occurred due to clinical application of the materials in light of the COVID-19 pandemic, can help the operator to reduce their working time, limit invasiveness and transmit a message of safety to the patient for the sustainable use of fast, effective products [36].

### 4.2. Salivary Demineralisation/Remineralisation

Regarding remineralisation in saliva, numerous studies have researched the role of saliva in the oral cavity, and implicitly, its properties on enamel. This topic is still debated in the literature, with saliva having new chapters to unravel. It is known that saliva can buffer and neutralise acids, a protective role; and is the first biological factor in preventing dental erosions and reducing demineralisation [37].

Lopes et al. demonstrated that the dental surface, after an acid attack, if exposed to saliva for some time before dental brushing, can be naturally remineralised [38]. The results of our study showed that the demineralisation of tooth enamel in artificial saliva with acidic pH 3 was initiated at 12 h and present after 48 h. However, in acidic saliva with pH 5, the lesion was initiated at 48 h. In the case of demineralised samples in saliva with pH 3, the remineralisation process occurred only after 72 h, while in the case of demineralised samples in saliva with pH 5, remineralisation was initiated at 24 h and was present at 72 h. This shows that the more acidic the pH is, the harder it is for saliva, alone, to achieve remineralisation. These results support other specific studies which demonstrated that the loss of dental structure becomes irreversible and more difficult to remineralise if exposure to acids is prolonged, and even increases, if abrasive factors are introduced [39,40,41].

However, it must be considered that the study used artificial saliva, which is known to have a higher mineral content and allows a higher deposition of minerals onto tooth enamel than human saliva. Artificial saliva has a higher time and potential for remineralisation, which means that the same in vitro study could have had different results if human saliva had been used.

Other studies conducted by Mutahar et al. in 2017, and Wetton in 2006, showed that salivary film offers protection against demineralisation [42,43].

As there are results in salivary remineralisation, both in our study and in the literature, it is very important that patients who show early signs of demineralisation are questioned about how they consume dietary acids. For prevention in these cases, it would be necessary to focus on diet and nutrition tips such as avoiding food acids between meals, reducing the consumption of carbonated drinks and, most importantly, recommending a diet rich in alkaline foods, which could increase the salivary pH. In order to demonstrate deficient eating habits, a very extensive study was carried out, which extended to seven countries in Europe and concluded that 1368 patients had a BEWE (basic erosive wear examination) score of 3, of which the factorial history was nutritional factors [44]. Moazzez et al., in their study, reported that patients with early demineralisation lesions consumed carbonated drinks in quantities higher than the control group, with the pH on the oral surface remaining low for longer [40].

Other specialised studies have concluded that saliva’s protective and remineralisation qualities are limited because the film functions more as a penetrable network to ions than as a barrier, and that salivary proteolytic enzymes and matrix metalloproteinases can contribute to the process of demineralisation lesion occurrence [45].

Aljulayfi et al., as in most of the studies mentioned above, analysed the rough surfaces profiled after demineralisation and exposition in saliva, and found that enamel surfaces no longer had the same roughness, but found it difficult to understand, only with the profile analysis, whether the action of saliva had a remineralisation or only a lubricating role [46].

Our samples were analysed with CLSM, both in-depth and on the surface, and the measurements showed that there was a possibility of remineralisation, but only after a long period of exposure to alkaline saliva with pH 8.

Regarding the influence of the acidic environment produced by demineralisation on the macro elemental composition of enamel, most studies refer to the impact on Ca ions. Analysis of the elemental composition of human enamel shows that Ca^2+^ accounts for ~37% of the mineral content by weight, with PO_4_^3−^ being the second most abundant at ~17% [36]. Enamel contains the highest Ca^2+^ content of all bioapatite. Compared to other tissues, enamel contains more than nine times the Ca^2+^ content of muscle or liver [47]. Most of the Ca^2+^ (~90%) found in enamel is incorporated during maturation [48,49]. Ca^2+^ uptake in enamel cells was, for many years, considered to be a passive phenomenon, with Ca^2+^ following a natural concentration gradient from the high levels found in interstitial fluid, to the lower concentration, in the cytosol [50,51].

There were studies focused on the impact of acidic beverages on Ca content after 12 h in different juices in the presence and absence of saliva. The results showed a decrease of around 30% of Ca concentrations in enamel, but the Ca loss was significantly reduced when saliva was used in acidic beverages. Ca loss in an acidic environment is due to the formation of complex compounds between Ca ions and citric acid, which increases dental minerals’ dissolution [43]. Some studies demonstrated that citric acid could produce more pronounced erosions than phosphoric acid at similar levels of acidity. Lussi et al., 2011, highlighted the potential of the remineralisation process to increase the Ca content on enamel surfaces [52].

Following our research, one outcome might be to create a scoring system [53] to assess and permit surveillance, evaluation and comparison of demineralisation/remineralisation in the clinical practice using a simplified digital workflow system, subjecting patients to the fewest possible appointments at the dentist [54]. Similar to other studies, the components could be identified clinically and assigned scores according to their statistical frequency, resulting in a system outcome score (SOS) [55].

## 5. Conclusions

Of the three treatments applied to the demineralised enamel samples, the most effective was the treatment with the low viscosity infiltrating resin Icon, from DMG, Hamburg, which fully penetrated the demineralisation lesion.

Fluoridation also further proved its importance, the results of fluoride treatment with Fluor Protector (Ivoclar Vivadent) being very effective in cases of medium-depth demineralised lesions.

Remineralisation of samples in artificial saliva was present, which is very important, but it is necessary to study remineralisation over a longer period, analysing the habits, diet and nutrition of patients in detail.

CLSM was a very effective tool in the detailed analysis of enamel; the laser measured the enamel surface and depth.

The demineralisation/remineralisation processes influenced the macro elemental composition of enamel demineralisation with natural saliva, which was less aggressive in decreasing Ca and Mg content.

## Figures and Tables

**Figure 1 materials-15-07258-f001:**
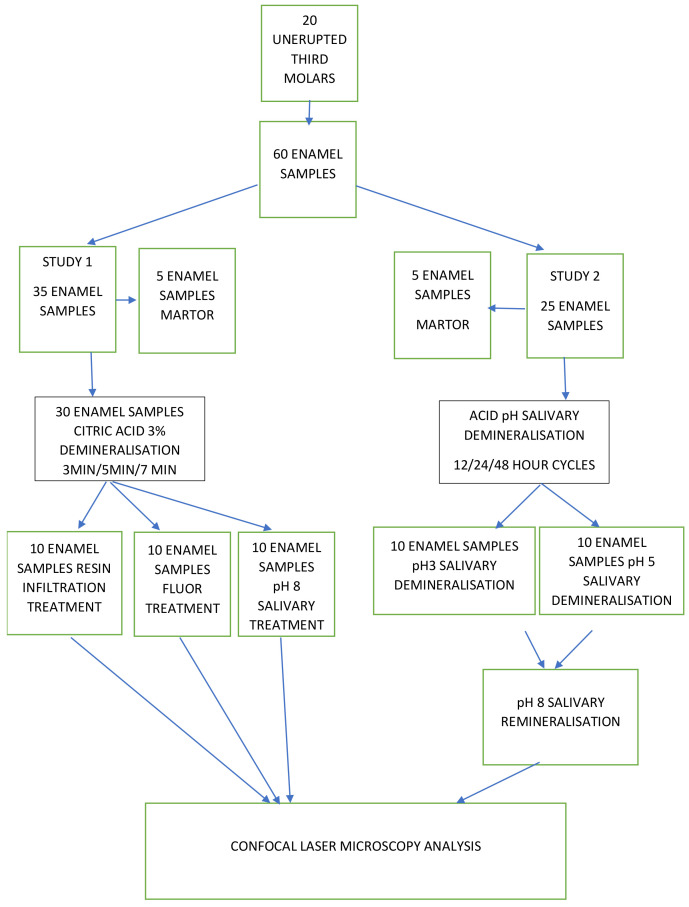
Diagram of the study protocol.

**Figure 2 materials-15-07258-f002:**
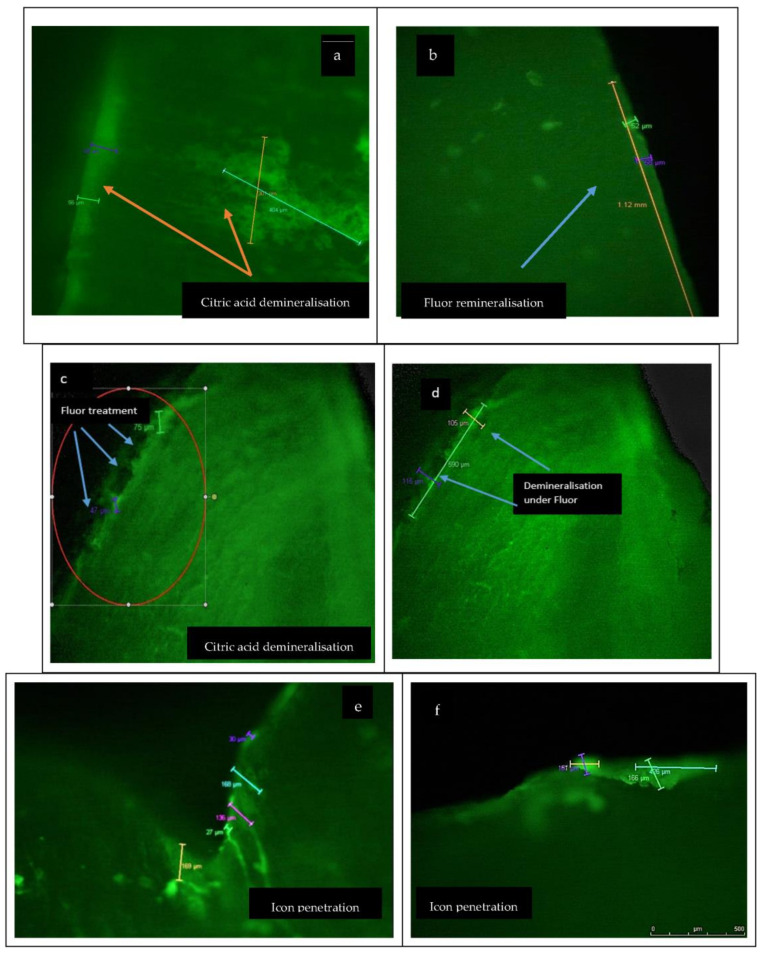
(**a**) CLSM of enamel demineralisation with 3% citric acid. (**b**) CLSM after Fluor Protector infiltration of enamel with 100% penetration (10× objective in dual fluorescence mode). (**c**) CLSM after Fluor Protector infiltration of enamel with partial penetration; (**d**) the lesions of demineralisation under Fluor Protector infiltration—partial penetration (10× objective in dual fluorescence mode); (**e**) CLSM after Icon infiltration of enamel; (**f**) Icon infiltration—depth of penetration; (**g**) the difference between Icon treatment and sound enamel; (**h**) 8 h box plot of laser penetration; (**i**) CLSM after pH 8 saliva remineralisation; (**j**) sound enamel samples shown by confocal laser microscopy analysis.

**Figure 3 materials-15-07258-f003:**
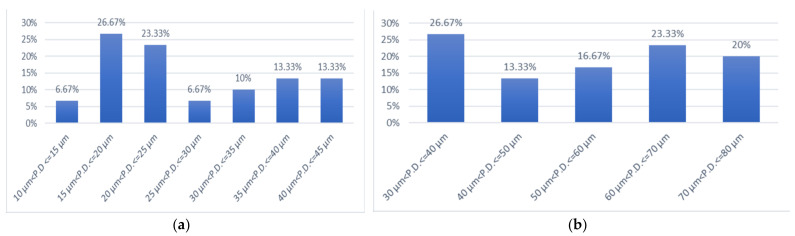
(**a**) Citric acid enamel demineralisation percentages after 3 min, at different penetration depths. (**b**) Citric acid enamel demineralisation percentages after 5 min, at different penetration depths. (**c**) Citric acid enamel demineralisation percentages after 7 min, at different penetration depths. P.D. means penetration depths.

**Figure 4 materials-15-07258-f004:**
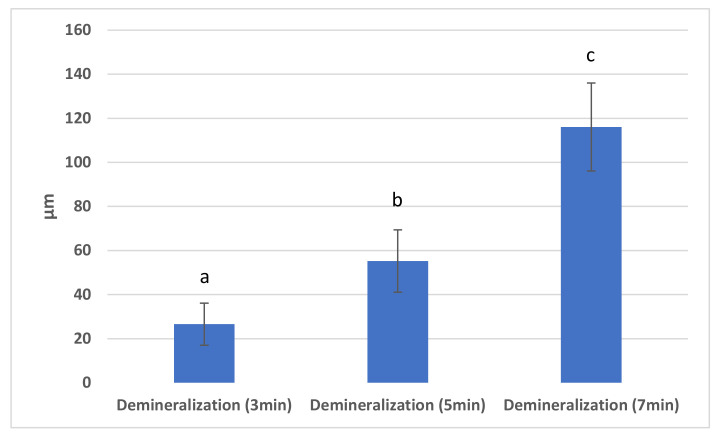
Demineralisation cycle. Different letters indicate statistically significant differences *p* ≤ 0.05 (Duncan’s test).

**Figure 5 materials-15-07258-f005:**
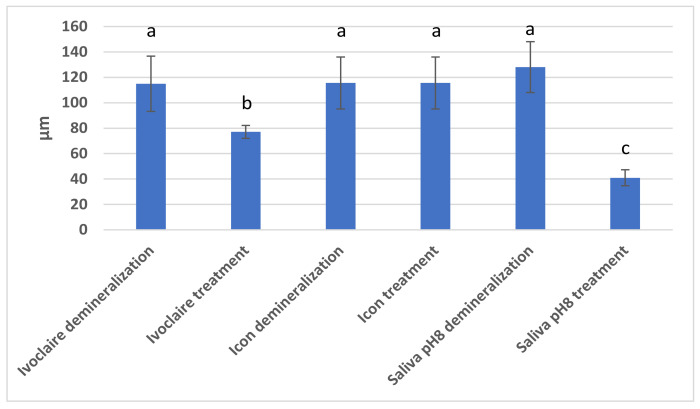
Demineralisation and Fluor Protector (Ivoclar Vivadent) treatment, resin infiltration treatment with Icon, (DMG Hamburg), and basic saliva pH treatment. Different letters indicate statistically significant differences *p* ≤ 0.05 (Duncan’s test).

**Figure 6 materials-15-07258-f006:**
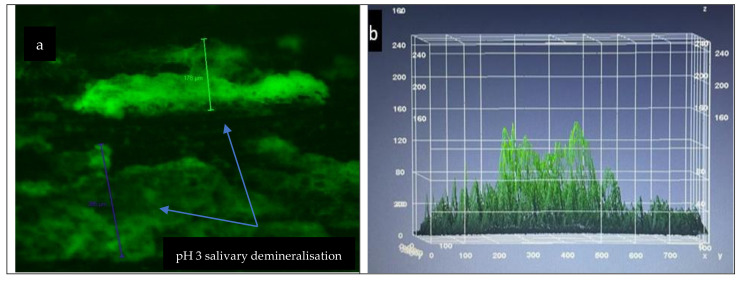
(**a**) pH 3 salivary demineralisation; (**b**) box plot of laser penetration.

**Figure 7 materials-15-07258-f007:**
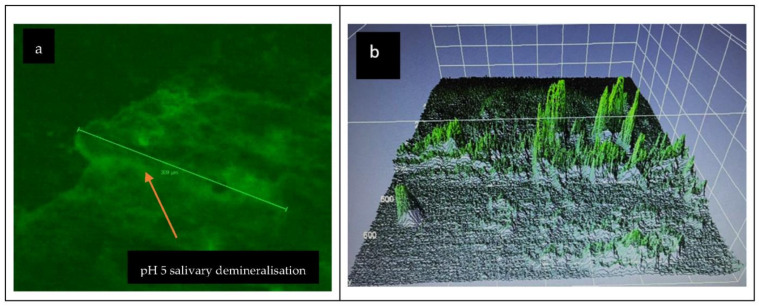
(**a**) pH 5 salivary demineralisation; (**b**) box plot of laser penetration.

**Figure 8 materials-15-07258-f008:**
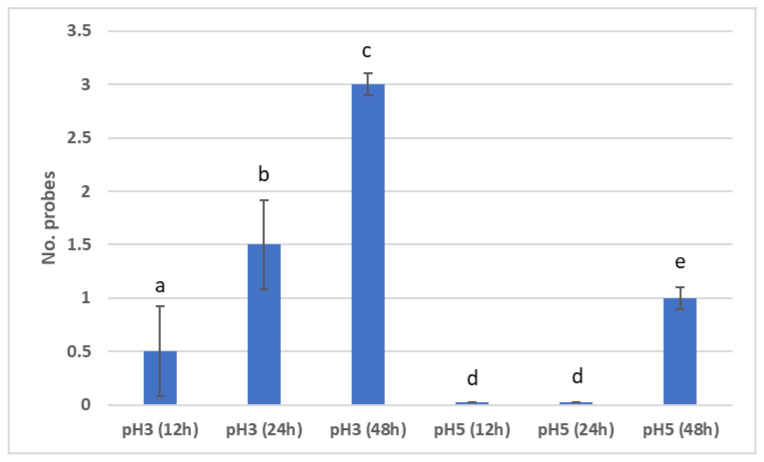
Demineralisation cycles: pH 3 and pH 5. Different letters indicate statistically significant differences *p* ≤ 0.05 (Duncan’s test).

**Figure 9 materials-15-07258-f009:**
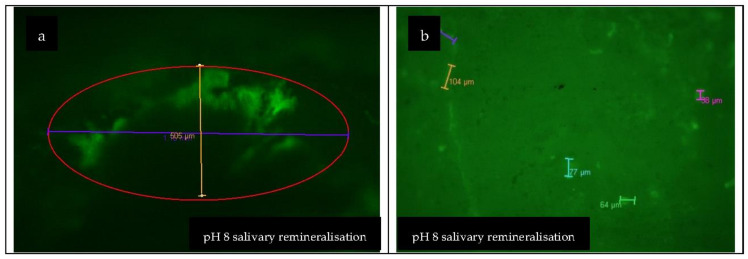
CLSM images of: (**a**) pH 8 salivary remineralisation; (**b**) pH 8 salivary remineralisation, magnified view.

**Figure 10 materials-15-07258-f010:**
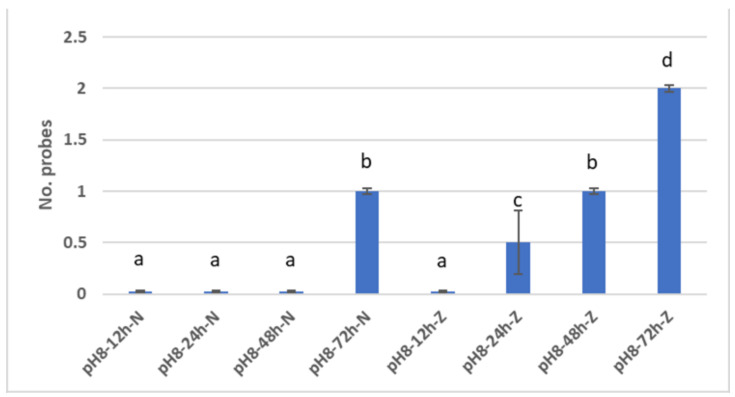
Remineralisation cycle: N and Z enamel samples. Different letters indicate statistically significant differences *p* ≤ 0.05 (Duncan’s test).

**Figure 11 materials-15-07258-f011:**
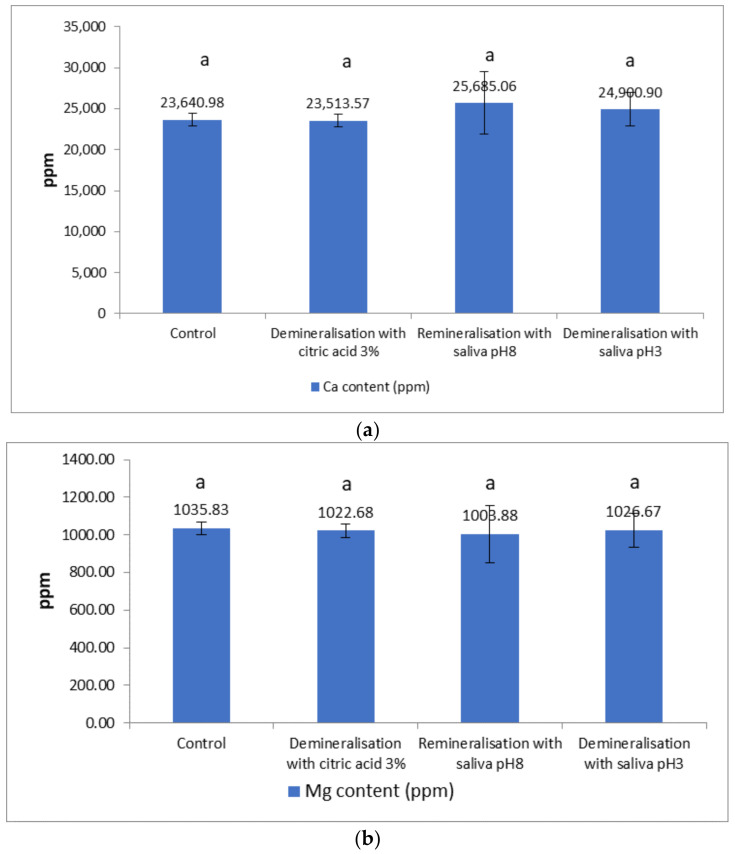
(**a**) Ca content (ppm) after demineralisation/remineralisation procedure; (**b**) Mg content (ppm) after demineralisation/remineralisation procedure; (**c**) K content (ppm) after demineralisation/remineralisation procedure. Different letters indicate statistically significant differences *p* ≤ 0.05 (Duncan’s test).

**Table 1 materials-15-07258-t001:** Protocol of in vitro demineralisation after CLSM analyses.

Citric Acid 3% at Room Temperature	3 min Immersion	5 min Immersion	7 min Immersion
Washing distilled water	30 s	30 s	30 s
Demineralisation lesion (CLSM)	Absence	Initiated	Present

**Table 2 materials-15-07258-t002:** The penetration depth of infiltration materials determined using CLSM (10× objective in dual fluorescence mode) before and after infiltration with chemical remineralisation agents.

Procedure	Penetration Depth (μm) Before Procedure		Penetration Depth (μm) After Procedure	
	Mean ± Std. Dev	Min.–Max Values	Mean ± Std. Dev	Min.–Max Values
Infiltration with Fluor Protector (Ivoclar Vivadent)	115 ± 21.72	87–143	77 ± 5.08	64–81
Infiltration with Icon resin infiltration (DMG)	115 ± 20.41	90–141	115 ± 20.41	90–141
Infiltration with artificial saliva with pH 8	118 ± 19.23	94–145	39 ± 5.99	30–45

## Data Availability

The report of the analyzes performed for the samples in the article can be found at the Interdisciplinary Research Platform (PCI) belonging to the University of Life Sciences “King Michael I” from Timisoara.

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
