# Peer review of "Impact of Dentistry Materials on Chemical Remineralisation/Infiltration versus Salivary Remineralisation of Enamel—In Vitro Study"

_materials, 2022, doi:10.3390/ma15207258_

Round 1
Reviewer 1 Report
This manuscript aimed to compare the chemical and natural remineralization of enamel under two demineralization protocols. The authors namely “natural remineralization” the protocol that used artificial saliva in the experiments. If the objective was to evaluate the natural process of enamel demineralization, in situ or in vivo experiments should be mandatory. It wasn’t clear what means chemical remineralization. The study lacks novelty. There are many studies that evaluated the penetration depth of resin infiltrant and the depth of remineralization with fluoride agents.
Abstract
Line 45 “iii) the demineralization/remineralization processes influences the macro elemental composition of enamel demineralization with natural saliva proving to be less aggressive in terms of decreasing Ca and Mg content.” Did the Ca and Mg content using saliva significantly lower than in other protocols? Please, give details about the AAS statistical results.
The research includes two directions: Study 1: Chemical demineralization versus chemical infiltration; Study 2: Natural demineralization versus natural infiltration. Why two demineralization protocols were used in the study?
Natural demineralization was made by keeping enamel samples in artificial acid saliva with pH 3 or pH 5. It should be better to submit specimens to des-re cycles, which better simulate the pH variations of the oral environment.
What about your sample size calculation? Please provide exact information, and add references
According to the title, the premise of the study was to evaluate the remineralization ability of dental materials. Icon is a resin-based material and doesn’t have the ability to release ions. Why it was used to evaluate the remineralization ability of dental materials?
Line 119. “The research analyzed 20 human teeth, with 60 enamel sections taken from unerupted wisdom teeth unaffected by caries.” What were the criteria to select the teeth? Did the age or mineral content of teeth standardize? The mineral content may influence the des-re process.
Line 146. Why the enamel samples were subjected to a micro abrasion protocol?
Table 2 could be removed from the manuscript and the protocol of treatment with Icon described in the text. There are repeated information in Table 2.
Line 275. “The results presented in the table 3 shown that before application the Fluor Protector-(Ivoclar Vivadent), the values recorded for the lesions of demineralized enamel samples were between 87 μm and 143 μm, with an average of 115 μm and decreased at 77 μm (minimum value 64 μm respectively 81 μm maximum value)” What means “decreased at 77 μm?
Line 285. “Table 3. The penetration depth of infiltration materials determined using CLSM (10× objective in dual fluorescence mode) before and after infiltration with chemical remineralization agents.” Add the means of demineralization depth and material infiltration.
2.5. Macro-elemental composition
To evaluate the macro-elemental composition of specimens, 3 g of sample was dissolved in the 6 N HCl and the macro-elements (Ca, K, Mg) were detected using Atomic Absorption Spectroscopy (AAS). In these experiments, both demineralized and sound enamel was used to obtain the specimens. This impairs the establishment of a relationship between remineralization protocols and the macro-elemental composition of the treated area.
It should be better to evaluate the mineral content in the region submitted to the different remineralization protocols. In this way, it wasn’t possible to evaluate the elements solely in the lesion.
Line 310. “Figure 3 presents citric acid enamel demineralization percentages, calculated according to formula (1), after 3 minutes (figure 3a), 5 minutes (figure 3b) and 7 minutes (figure 3c) of exposure.” Where is the formula?
Line 373. “Opposite, no statistical significant differences were observed between the Icon demineralized sample and the sample with Icon treatment. These results highligthed the ca”. There is missing content in the sentence.
Line 385. “Of the three treatments applied to the samples of enamel demineralised in citric acid, the most effective was the treatment with infiltrating resin with low viscosity Icon from DMG Hamburg, which managed to penetrate the demineralisation lesion fully. Fluoridation also further proves its importance, the results of fluorine treatment with Ivoclar Vivadent being excellent but unable to equal the infiltrating resin. Remineralization of samples in artificial saliva was present, which is very important, but here it is necessary to study remineralization over a more extended period.” Figure 5 shows a reduction in the depth of demineralization for Fluor Protector and saliva treatment. Why Icon was considered the most effective treatment? Icon can’t remineralize enamel, but only fill pores.
Line 400. “After CLSM analyzing of the surfaces of the enamel samples, (Figure 6), it can be observed that the samples introduced into the artificial saliva with pH 3 show incipient demineralization lesions at 12 hours after immersion, medium lesions at 24 hours after immersion and present evident lesions, more extensive in the surface at 48 hours.” How long the specimens of figure 6 were soaked in saliva?
Author Response
Date: Timisoara /11.09.2022
Name: Diana Obistioiu
University: Banat’s University of Agricultural Sciences and Veterinary Medicine, „King Michael I of Romania’’ from Timisoara, Calea Aradului No.119, 300645, Timisoara, Romania
Address: Calea Aradului No. 119, 300641 Timisoara, Romania
E-mail: dianaobistioiu@usab-tm.ro
Rebuttal letter
Dear Editor
We would like to address all our thanks and gratitude for the constructive observations, corrections and recommendations.
Based on the reviewers’ recommendations, the authors of this paper responded point by point to the following aspects:
- Reviewer 1 comments:
This manuscript aimed to compare the chemical and natural remineralisation of enamel under two demineralisation protocols. The authors namely “natural remineralisation” the protocol that used artificial saliva in the experiments. If the objective was to evaluate the natural process of enamel demineralisation, in situ or in vivo experiments should be mandatory. It wasn’t clear what means chemical remineralisation. The study lacks novelty. There are many studies that evaluated the penetration depth of resin infiltrant and the depth of remineralisation with fluoride agents.
Because our study is an in vitro one, in order not to create confusion and misunderstandings, we will modify the natural remineralisation to the salivary remineralisation. Also, in order to avoid confusion between resin infiltrating and remineralising therapy, we suggest a change in the title Impact of Dentistry Materials on Chemical Remineralization/Infiltration versus salivary Remineralisation of Enamel-in vitro study.
You are right, there are studies that have evaluated the penetration of infiltrating resins and the penetration of flour, but we want to emphasise the importance of the results of this study for the clinical practice of the dentist: because in clinical practice we meet with many cases of lesions of enamel demineralisation, it is very important that the dentist, after correctly diagnosing the type of lesion, can also choose the therapy suitable for his case:
- For incipient enamel lesions, it can only be recommended to modify the diet to increase the salivary pH, and remineralisation to be a natural process with the help of saliva;
- If the diagnosis is of medium demineralisation lesion, one can choose as a treatment based on fluoride solutions, the study showing that in this case fluoride therapy has very good results.
- When the demineralisation lesions are of enamel lesion type - (without loss of substance) the remineralising fluoride therapy is no longer sufficient, in this case it is necessary to infiltrate the enamel with resin with low viscosity to stop the demineralisation process by obliteration of the pores. The diagnosis of demineralisation can be accurately performed in clinical practice with the help of the DiagnoDent laser diode.
Abstract
Comment: Line 45 “iii) the demineralisation/remineralisation processes influences the macro elemental composition of enamel demineralisation with natural saliva proving to be less aggressive in terms of decreasing Ca and Mg content.” Did the Ca and Mg content using saliva significantly lower than in other protocols? Please, give details about the AAS statistical results.
Answer: The explanation and modifications were inserted in the text.
Comment: The research includes two directions: Study 1: Chemical demineralisation versus chemical infiltration; Study 2: Natural demineralisation versus natural infiltration. Why two demineralisation protocols were used in the study?
Answer: This study includes two protocols of demineralisation because in the oral cavity, several types of agents produce the demineralisation lesion, namely: carbonated drinks, carbohydrate consumption, different acid gels applicated on enamel, gastroesophageal reflux disease, acid pH of saliva. For our study, we chose citric acid with a pH similar to that of carbonated drinks, which has a short but aggressive demineralisation action and a demineralisation protocol with acid saliva, which can simulate oesophagal reflux disease or the acid salivary pH of a patient with a balanced diet, which is not so aggressive at the moment on enamel but can cause damage in a continuous process.
Comment: Natural demineralisation was made by keeping enamel samples in artificial acid saliva with pH 3 or pH 5. It should be better to submit specimens to des-re cycles, which better simulate the pH variations of the oral environment.
Answer: The protocol was established according to other studies, the chosen variables being the periods selected (24, 48 hours).
Comment: What about your sample size calculation? Please provide exact information, and add references
Answer: Given the similar articles attached to the bibliography, where some authors had fewer enamel samples than those present in the study, and others more, we considered that our number of 60 samples from 20 human teeth, third permanent molars from patients older than 18 years, is relevant to this study.
According to the references W.H.Arnold Et al had in their study 12 extracted caries free human incisors; R.Perey et al : 24 fluorosed human molars and premolars; Elsami B. et al : 15 pecients who needed 2 premolars extracted; Hendrik Meyer-Luekel et al : 20 extracted permanent human posterior teeth non cavitated caries; Arnold W.H. et al : 28 permanent teeth extracted with non cavitated caries lesion; Enas T.Enan et al: 45 extracted premolars giving 90 specimens; Krunal Chokshi et al : 60 enamel specimen; Jing Yhang et al – 50 artificial enamel white spots.
Comment: According to the title, the premise of the study was to evaluate the remineralisation ability of dental materials. Icon is a resin-based material and doesn’t have the ability to release ions. Why it was used to evaluate the remineralisation ability of dental materials?
Thank you for your notification. Indeed, it creates confusion in the title, which is why we suggest modifying it. The icon was used in the study as infiltrating therapy to show that when fluoride on the demineralisation lesions no longer penetrates enough to produce the remineralisation of the entire lesion, because the lesion is too deep, to prevent its transformation from a demineralisation lesion into a carious one, one can choose the infiltrating therapy with resin with low viscosity that manages to penetrate very much in depth.
Comment: Line 119. “The research analysed 20 human teeth, with 60 enamel sections taken from unerupted wisdom teeth unaffected by caries.” What were the criteria to select the teeth? Did the age or mineral content of teeth standardise? The mineral content may influence the des-re process.
Answer: Orthodontically extracted teeth from patients over 18 years of age who had not been subjected to any influence from the oral cavity were chosen for this study. We did not measure or standardise the degree of mineralisation of the teeth, as we took the average demineralisation of each individual sample as a starting point to follow the evolution of the demineralisation/remineralisation process and infiltrative therapy.
Comment: Line 146. Why the enamel samples were subjected to a micro abrasion protocol?
Answer: Because the enamel samples were taken from unerupted, intact molars that had not been subjected to any abrasion in the oral cavity, we tried to simulate as closely as possible the environment in the oral cavity.
Comment: Table 2 could be removed from the manuscript and the protocol of treatment with Icon described in the text. There are repeated information in Table 2.
Answer: The modifications were made.
Comment: Line 275. “The results presented in the table 3 shown that before application the Fluor Protector-(Ivoclar Vivadent), the values recorded for the lesions of demineralised enamel samples were between 87 μm and 143 μm, with an average of 115 μm and decreased at 77 μm (minimum value 64 μm respectively 81 μm maximum value)” What means “decreased at 77 μm?
Answer: The demineralisation lesion measured in depth before fluoride therapy had an average of 115 μm, which after fluoride therapy decreased to the value of 77μm, which means that remineralisation of a portion of the lesion occurred.
Comment: Line 285. “Table 3. The penetration depth of infiltration materials determined using CLSM (10× objective in dual fluorescence mode) before and after infiltration with chemical remineralisation agents.” Add the means of demineralisation depth and material infiltration.
Answer: Thank You, the modifications were inserted in the manuscript.
Comment: 2.5. Macro-elemental composition
To evaluate the macro-elemental composition of specimens, 3 g of sample was dissolved in the 6 N HCl and the macro-elements (Ca, K, Mg) were detected using Atomic Absorption Spectroscopy (AAS). In these experiments, both demineralised and sound enamel was used to obtain the specimens. This impairs the establishment of a relationship between remineralisation protocols and the macro-elemental composition of the treated area.
Answer: The samples used in the macro-elemental composition belong to the same samples divided into equal parts, and the entire surface of the selected sample was emersed according to the protocol in the de/remineralising solution; therefore, the entire sample may be analysed by AAS and the data used for statistical analysis. The results are expressed as a report of the value obtained divided by the mass taken into analysis before acid immersion, so the values obtained are easily extrapolated.
Comment: It should be better to evaluate the mineral content in the region submitted to the different remineralisation protocols. This way, it wasn’t possible to evaluate the elements solely in the lesion.
Answer: This issue was also a problem we tried to resolve by emerging the entire sample surface according to protocol, therefore not having lesions smaller than the entire surface taken into analysis.
Comment: Line 310. “Figure 3 presents citric acid enamel demineralisation percentages, calculated according to formula (1), after 3 minutes (figure 3a), 5 minutes (figure 3b) and 7 minutes (figure 3c) of exposure.” Where is the formula?
Answer: The formula and modifications were inserted in the text.
Comment: Line 373. “Opposite, no statistical significant differences were observed between the Icon demineralised sample and the sample with Icon treatment. These results highligthed the ca”. There is missing content in the sentence.
Answer: There was a text error. Thank you, the corrections were made.
Comment: Line 385. “Of the three treatments applied to the samples of enamel demineralised in citric acid, the most effective was the treatment with infiltrating resin with low viscosity Icon from DMG Hamburg, which managed to penetrate the demineralisation lesion fully. Fluoridation also further proves its importance, the results of fluorine treatment with Ivoclar Vivadent being excellent but unable to equal the infiltrating resin. Remineralisation of samples in artificial saliva was present, which is very important, but here it is necessary to study remineralisation over a more extended period.” Figure 5 shows a reduction in the depth of demineralisation for Fluor Protector and saliva treatment. Why Icon was considered the most effective treatment? Icon can’t remineralise enamel, but only fill pores.
Answer: CLMS measurements have the advantage of measuring much deeper, and our results showed that in terms of remineralisation, fluoride has a limited ability to penetrate deep, and if the lesion is much deeper, there will remain a demineralised area below that was penetrated by fluoride. On the other hand, Icon manages to penetrate deep lesions, obliterating the pores. This is why this comparison is important so that the dentist knows how to optimally choose in clinical practice the therapy he will apply to the demineralisation lesion: remineralisation with Fluoride and increasing the salivary pH or infiltration therapy with low viscosity resins.
Comment: Line 400. “After CLSM analysing of the surfaces of the enamel samples, (Figure 6), it can be observed that the samples introduced into the artificial saliva with pH 3 show incipient demineralisation lesions at 12 hours after immersion, medium lesions at 24 hours after immersion and present evident lesions, more extensive in the surface at 48 hours.” How long the specimens of figure 6 were soaked in saliva?
Answer: Figure 6 shows enamel demineralisation lesions after 48 hours of immersion in acid saliva with pH 3. The explanation was inserted in the text.
The manuscript was corrected from the point of view of grammatical errors, and the corrections were made in the text.
Sincerely,
Dr. Obistioiu Diana

Reviewer 2 Report
The paper by Damian et al. examined the enamel changes using Confocal Laser Scanning Microscopy (CLSM), along with the calculation of the depth of lesions and the demineralization/remineralization percentage. As I am not an expert for this special area, I can only raise few questions, which may help improve this paper.
1. Do you need ethic approval to study or using third molar samples, which I guess not.
2. Just wondering if the patients age or gender difference may have any impact on the result?
3. As human saliva or mouth contains millions of microorganisms, do you think your artificial saliva will simulate the actual situation in human mouth?
4. Do you think using SEM may help you to get better and clear images to facilitate your study?
5. In figure 2a, why do you think the bright green parts are citric acid demineralization? Is this based on ay reference or other quantification?
6. Figure 3a-c have no error bar.
7. The author better keeps the same style for making the figures, while Figure3a-c and figure4&5 use another style.
8. Figure4&5 the x-axis angled at different direction.
9. There are no scale bars in all your confocal images. The quality of the confocal images are not really in good quality. Leica TCS SPE is a very good confocal laser scanning microscope. The author should capture very good quality using this machine. I also think the other reason is the magnification is too low, especially if the author wants to take images showing details.
10. I am not sure if this article fits into the aim and scope of the Journal. Please double check.
Author Response
Date: Timisoara /11.09.2022
Name: Diana Obistioiu
University: Banat’s University of Agricultural Sciences and Veterinary Medicine, „King Michael I of Romania’’ from Timisoara, Calea Aradului No.119, 300645, Timisoara, Romania
Address: Calea Aradului No. 119, 300641 Timisoara, Romania
E-mail: dianaobistioiu@usab-tm.ro
Rebuttal letter
Dear Editor
We would like to address all our thanks and gratitude for the constructive observations, corrections and recommendations.
Based on the reviewers’ recommendations, the authors of this paper responded point by point to the following aspects:
- Reviewer 2 comments:
The paper by Damian et al. examined the enamel changes using Confocal Laser Scanning Microscopy (CLSM), along with the calculation of the depth of lesions and the demineralisation/remineralisation percentage. As I am not an expert for this special area, I can only raise few questions, which may help improve this paper.
Answer: We greatly appreciate your time and input in improving our manuscript.
Comment: Do you need ethic approval to study or using third molar samples, which I guess not.
Answer: We had ethical approval for our study since the samples weretaken inside a private practice and the data used in a PhD thesis but did not think we needed to insert it in the manuscript.
We have an agreement signed by the patient that the extracted tooth can be used for research purposes. We also have the ethical approval of the SCIENTIFIC RESEARCH ETHICS COMMITTEE Nr. 45/28.09.2018 of Victor Babes University.
Thank You for your kind suggestion; the data was inserted in the paper.
Comment: Just wondering if the patients age or gender difference may have any impact on the result?
Answer: No, since once maturity is reached, the mineralisation of an adult tooth is ended, and the age difference is no longer important. Differences may occur if the tooth is taken from smaller children under 18. Gender does not affect the results also since the comparison is made between the same samples divided for different activities within our research.
Comment: As human saliva or mouth contains millions of microorganisms, do you think your artificial saliva will simulate the actual situation in human mouth?
Answer: Yes, in the case of our study, it does because the microorganisms in the oral cavity have no part in the mineralisation of the tooth. Once maturity is reached, the mineralisation is ended, and the microorganisms work more in plaque forming or biofilm appearance on the tooth surface.
Saliva was used in this study only in terms of Ph+, being interested in its acid-base properties, not in microorganisms. In vivo, pH triggers the process of demineralisation/remineralisation in the oral cavity by migrating ions from enamel into saliva and vice versa.
We have initiated demineralisation without considering the oral cavity's microorganisms. Therefore we suggest changing the title from natural remineralization to salivary remineralization to avoid confusion.
Comment: Do you think using SEM may help you to get better and clear images to facilitate your study?
Answer: 5. Studies use SEM for enamel analysis but evaluate the enamel's surface area and microhardness. SEM cannot reveal the degree of demineralisation in-depth, so we considered it opportune to see the depth at which the demineralisation occurs and what is the best way of action for remineralisation, for which we considered the confocal microscope better suited.
Comment: In figure 2a, why do you think the bright green parts are citric acid demineralisation? Is this based on ay reference or other quantification?
Answer: Initially, as a preliminary analysis, intact enamel samples were analysed, not having undergone any demineralisation process; after what time, the samples were immersed in acid citric and then reanalysed, where differences in enamel structure could be observed. The procedure was done previously in the research laboratory as presented in the literature cited [12].
Comment: Figure 3a-c have no error bar.
Answer: Thank You, the modifications were made.
Comment: The author better keeps the same style for making the figures, while Figure3a-c and figure4&5 use another style.
Answer: The modifications were inserted.
Comment: Figure4&5 the x-axis angled at different direction.
Answer: The changes were done.
Comment: There are no scale bars in all your confocal images. The quality of the confocal images are not really in good quality. Leica TCS SPE is a very good confocal laser scanning microscope. The author should capture very good quality using this machine. I also think the other reason is the magnification is too low, especially if the author wants to take images showing details.
Answer: Yes, You are right. The research took place in an educational institution where bureaucracy hinders software improvement. We hope this will be solved in the future, and the following articles will have higher-quality images. Thank You for understanding this issue.
Comment: I am not sure if this article fits into the aim and scope of the Journal. Please double check.
Answer: We submited the article after a close analysis and found several other similar articles published within Materials. We can cite here some of them:
Cao, C.Y.; Mei, M.L.; Li, Q.-l.; Lo, E.C.M.; Chu, C.H. Methods for Biomimetic Mineralisation of Human Enamel: A Systematic Review. Materials 2015, 8, 2873-2886. https://doi.org/10.3390/ma8062873
Prodan, D.; Moldovan, M.; Chisnoiu, A.M.; SaroÈ™i, C.; Cuc, S.; Filip, M.; Gheorghe, G.F.; Chisnoiu, R.M.; Furtos, G.; Cojocaru, I.; Delean, A.G.; Cimpean, S.I. Development of New Experimental Dental Enamel Resin Infiltrants—Synthesis and Characterization. Materials 2022, 15, 803. https://doi.org/10.3390/ma15030803
Paolone, G.; Moratti, E.; Goracci, C.; Gherlone, E.; Vichi, A. Effect of Finishing Systems on Surface Roughness and Gloss of Full-Body Bulk-Fill Resin Composites. Materials 2020, 13, 5657. https://doi.org/10.3390/ma13245657
Answer: Thank you for the appreciation.The entire manuscript has been carefully checked and corrected.
The manuscript was corrected from the point of view of grammatical errors, and the corrections were made in the text.
Sincerely,
Dr. Obistioiu Diana

Round 2
Reviewer 2 Report
I have no further comments. Thanks.
Author Response
Thank you very much for all the constructive suggestions that helped to improve the article in terms of form and content.